# “It Was Just Really Hard to Be Pregnant in a Smaller Town …”: Pregnant and Parenting Teenagers’ Perspectives of Social Support in Their Rural Communities

**DOI:** 10.3390/ijerph192416906

**Published:** 2022-12-16

**Authors:** Lauren Baney, Alison Greene, Catherine Sherwood-Laughlin, Jonathon Beckmeyer, Brandon L. Crawford, Frederica Jackson, Lisa Greathouse, Dechen Sangmo, Michaella Ward, Susan Kavaya

**Affiliations:** 1School of Public Health, Indiana University-Bloomington, Bloomington, IN 47405, USA; 2College of Applied Human Sciences, West Virginia University, Morgantown, WV 26506, USA; 3Community Health, Indiana University Health, Bloomington, IN 47401, USA

**Keywords:** teenage pregnancy, rural communities, qualitative research, social support, emotional support

## Abstract

Teenage pregnancy has a history of being a “social problem” in the United States, with there being higher rates in rural communities. Social support, a contributor to improving mental health outcomes, can significantly impact a teenager’s pregnancy and parenting experience. Using House’s (1981) social support framework, this study explores the teenagers’ perceptions of how their rural community reacted and responded to them as pregnant and parenting teenagers. The results were formulated through the thematic analysis of semi-structured interviews (n = 26) with current and former pregnant and/or parenting teenagers. The participants reported experiencing both positive and negative social support. There were more reports of emotional support and instrumental support among the forms of positive support than there were in the other categories. Informational support was lacking. The appraisal support from community members was negative. There is a need for rural communities to develop effective social support strategies to provide positive support for pregnant and parenting teenagers.

## 1. Introduction

In 2021, the United States had a birth rate of 14.4 births per 1000 15–19-year-old women [1]. The current teenage birth rate is due to its steady decline since the peak teenage birth rate in the early 1900s [2]. Although teenage birth has declined in all areas of the Unites States, the decline has been greatest in urban communities [2]. The slower decline in teenage birth rates in rural communities is due to them having fewer socioeconomic resources and opportunities [3]. For instance, rural teenagers typically have few opportunities to receive sexual health education and have limited sexual health and reproductive health education [4,5]. Rural communities may also have limited reproductive healthcare services, and teenagers may not feel comfortable utilizing the services that are available [3,6,7]

Pregnant and parenting teenagers’ discomfort utilizing the available healthcare services is likely a reflection of the stigma that continues to surround teenage pregnancy and parenthood in rural communities [8]. This stigma can become a barrier to rural healthcare access, and it may manifest itself in the pregnant and parenting teenagers’ experiences of negative interactions with healthcare providers particularly in the healthcare settings [8,9,10,11,12]. For example, Craig and Stanley (2006) found that rural pregnant teenagers felt unable to escape scrutiny and stigma from healthcare providers and pharmacists due to a lack of choice in the available healthcare resources. Rural pregnant and parenting teens believe they are highly visibility in the community, perceiving that everybody knows everybody in the rural communities, and the state of pregnancy elicits negative attention from other community members [8]. Kessler et al. (2018) found that females when they are compared to males tended to have a more difficult time accessing healthcare due to stigma. Teenage pregnancy has been viewed as a “women’s problem”, as females typically carry much of the burden of stigma and responsibility for the pregnancy and care of the child than their male counterparts do [13].

Rural pregnant and parenting teenagers need meaningful social support as they navigate the experiences and related decisions of pregnancy and parenting decisions, particularly in stigmatized settings. Social support may be especially important when they are utilizing healthcare resources for themselves and their children. Therefore, understanding where rural pregnant and parenting teenagers find and use social support has important implications for public health prevention and intervention efforts aimed at supporting the well-being of pregnant and parenting teens and their children.

### Conceptual Framework

The present study is based on the conceptual framework of social support which is defined as an individual’s perception and actuality that one is cared for and has available assistance from others and that an individual is part of a supportive social network (House, 1981). Social support can be separated into four forms (see Figure 1): (1) emotional, (2) instrumental, (3) informational, and (4) appraisal [14]. Emotional support refers to having a reliable alliance that is built on companionship, affection, and intimacy [15]. Instrumental support is defined by having tangible assistance, such as with finances, tasks, and services. Informational support is defined by guidance and advice [16]. Appraisal support is defined as the communication of expectations and feedback from others, which additionally contributes to self-esteem [17,18].

Social support is an integral part of mitigating stress and combating dire health outcomes during an unplanned pregnancy [19] and it has the potential to improve the health outcomes across a broad range of socially effective health areas, such as mental health and anxiety [20]. One study reported that it was the quality of relationships and an increased number of support networks that offered crucial support to teenagers during pregnancy [21]. Renbarger and colleagues used House’s social support theory in a small-scale study to test if all four forms of social support enhanced their health outcomes. They found the maternal and child health outcomes may be improved through providing social support by helping to promote positive and trust-based relationships between the women and their health care providers during the prenatal care process [22] Studies have also suggested that there needs to be more focus on social support for parenting adolescents, as they are still children themselves, and the parenting aspect may be left out after the pregnancy [23] In the present study, we explored the pregnant teenagers, teenage parents, and parents who had children as teenagers’ perceptions of the social support they received in their rural community. By documenting where rural pregnant and parenting teens receive social support from and how it has helped them navigate pregnancy and parenting decisions, we aim to inform public health resources for pregnant and parenting teens and their children.

## 2. Materials and Methods

The present study is based on data from the Project Uncovering New Initiative for Teen Pregnancy (UNITE). Project UNITE was a community–academic partnership consisting of a researcher group from a University in a Midwest state and community organizations in two rural counties in that state. The overall project team included undergraduate and graduate students, faculty members, community health professionals, and community leaders from both of the partnering counties. The present research was carried out by graduate students and faculty members from the project team. Both of the counties had teenage birth rates that were higher than the state average. Project UNITE utilized a community-based participatory research approach to identify the conditions that may be contributing to early pregnancy and parenthood and to work with community leaders to develop community-based prevention and intervention strategies. The study was funded by a center that addresses the challenges in the Midwest state’s rural communities’ experience and enhances the opportunities in collaboration with those communities. The study received IRB approval from the Human Subjects Office on the University’s campus. The current study focuses on teenage pregnancy and parenting experience from the perspective of current and former pregnant and/or parenting teenagers and uses a subset of data from a larger study.

### 2.1. Procedures

Semi-structured qualitative interviews were conducted with current and formerly pregnant and/or parenting teenagers from two rural counties in a Midwest state. We allowed both the current and former teen mothers and fathers to participate in our project. The participants were recruited in several ways. Firstly, members of the community coalitions focused on reducing teenage pregnancy, social service agencies, schools, healthcare providers, and faith-based organizations were asked to nominate current and former teen parents to participate in our interviews. We contacted each person who was nominated and invited them to participate in our project. Secondly, an obstetrician sent letters to the current and recently pregnant teens. We also placed flyers advertising the opportunity to participate in interviews in social service and healthcare offices. Finally, at the conclusion of each interview, we asked the participants if they knew anyone that they thought we should invite to participate in an interview. The interviews were conducted by the research assistants (the majority of whom were graduate students), faculty members, and a one research partner serving on the internal research team.

Prior to each interview, the participants provided verbal consent to be interviewed and audio recorded. In order to build trust and rapport, all of the participants were informed about who referred them for the interview and the local coalition with whom we partnered. Additionally, we established trust with the participants by having the first interview questions focused on their personal experiences and observations of living in their respected counties. The interviews followed a guide that included prompts for a range of topics including experiences with pregnancy and parenthood, how they perceived their communities, resources within their communities, perceptions of community attitudes and values, and suggestions for future resources, programs, or interventions. (See Appendix A for copy of the interview guide) The interview duration spanned 45–60 min. All of the interviews took place between March 2020 and May 2021. Due to the COVID-19 restrictions in place at the time of the data collection, all of the interviews were conducted by telephone. The participants received a USD 25 gift card for participating in the study.

### 2.2. Data Analysis

The interviews were recorded and transcribed using a confidential transcription service. One research assistant was responsible for uploading each interview recording to the transcription service. The same research assistants who conducted interviews also read and cleaned the interview transcripts by correcting the transcription errors and de-identifying the participant information. The data analysis followed the steps prescribed by Braun and Clark (2006) for conducting thematic analysis. The coding was carried out using Atlas.ti software [24]. According to Braun and Clark (2006), the purpose of a thematic analysis is to identify, analyze, and report patterns within data. In our initial review of the interview transcripts, a theme of social support began to emerge. Drawing on House’s (1981) model, we developed a broad conceptualization of social support which involved how the networks, which included members such as family, friends, teachers, healthcare professionals, and other community members, offered care, guidance, information, or empathy towards a pregnant or parenting teenager. One research assistant then coded all of the interviews for data pertaining to the current and formerly pregnant and parenting teenagers’ experiences with the social support (or the absence of social support) related to pregnancy and parenthood. The research assistant shared their coding with the larger research team to prevent subjectivity issues. Once they were agreed upon, the data were extracted from the interviews, and then the same research assistant conducted a second analysis that was guided by House’s (1981) four aspects of social support: (1) emotional support, (2) instrumental support, (3) informational support, and (4) appraisal support. The social support data were grouped into those four areas, and we identified more specific themes within each area to illustrate the forms and functions of social support within our participants experiences of teenage pregnancy and parenthood.

## 3. Results

A total of 26 current and formerly pregnant and/or parenting teenagers participated in this study. All of the participants lived in the partnering counties during their pregnancy and became pregnant or parents at the ages of 15–19 years old. Four participants were currently pregnant teenagers at the time of the interview, and they were all 18 or 19 years old. Twenty-two participants had children as teenagers and were over 19 years old at the time of the interview. Most of the participants identified as female (*n* = 23), and three participants of them identified as male.

Overall, these pregnant teenagers and parents who had had children as teenagers, described multiple forms of social support. Emotional and instrumental support were recalled more frequently than informational or appraisal support were. Further, our participants did not find actual support within each of House’s (1981) domains of social support. Within the appraisal domain, the community members were more judgmental than they were supportive. The participants’ experiences within each social support domain are provided below.

### 3.1. Emotional Social Support

Emotional support is driven by trust, empathy, and care. The participants shared experiences of how the individuals (e.g., parents, church members, and peers) or social service organizations provided emotional support by showing care and empathy during their pregnancy or while they were parenting. The participants often discussed emotional support within the context of important rituals associated with pregnancy and parenthood, such as baby showers.


*“I never once felt shamed by my church. They threw a wonderful -- my best friend and the church helped throw me a wonderful baby shower. So, I think there’s an opportunity for the church to step up, the churches in our local areas to step up and welcome, you know, like, be there for the teen pregnant women.”*

*- Former teenage parent*



*“My mother-in-law was probably my biggest support for all of that. Helped me with the baby whenever I needed to take a shower or if I needed to eat. If I needed just a nap for a minute, she’d take him for me, and that was a big help. My fiancé did a lot of the same, and his grandma, my fiancé’s grandma, always came over. She was over here at least once a week making sure that we had everything that we needed, making sure to give me a minute. She was a real sweet woman. I love her to death. She still comes and sees us as often as she can. With everything going on right now, she hasn’t been able to because her husband has heart disease, but it’s definitely -- I definitely have a really good support system now. And they’ve helped out a lot.”*

*- Current teenage parent*


These events appeared to be important demonstrations that family members, friends, and social networks wanted to help during their transition to parenthood. Thus, tangible experiences can foster the perceptions of a positive support system.

However, not all of the participants received the emotional support they needed during their pregnancy and when they were adjusting to parenting. In some cases, previously close relationships with parents and friends worsened due to their pregnancy. When important relationships lacked emotional support, the pregnant and parenting teens felt more stress in their lives.


*“Because whenever you become a teen parent, a lot of times you lose a lot of your support system from friends, from family. Around here there’s a lot of very conservative families, and it’s really hard to keep some of your friends after that. And in my personal case, I got pregnant and probably lost almost every single one of my friends because their parents didn’t want them to talk to me anymore. So whenever you lose that support system, sometimes it makes it a lot harder to get things done, and it can be stressful. It’s hard to stick with the goals that you have set, and it’s harder to get them done when you wanted to get them done at.”*

*- Current teenage parent*


For some of the participants, their family members may not have initially provided emotional support, but over time they began to accept the pregnancy, leading to care and emotional support.


*“Well, my mother disowned me for two weeks. Because they found out because I was going up to swear into the Army and found out I was pregnant, so my mother did not respond to well that there was the second teenage pregnancy in our family. We are a mixed, we are a blended family. So, I got disowned for two weeks…and my mom eventually did come around to and was supportive and helpful.”*

*- Former teenage parent*


### 3.2. Instrumental Support

Instrumental support involves providing tangible services that support the teenagers during pregnancy and parenting. For pregnant and parenting teens, instrumental support is critical for reducing (or removing) the barriers to completing important developmental tasks such as completing their educations. For example, our participants shared how people within their school systems made it possible for them to continue attending school and earn their diplomas.


*“Being in [redacted name] class while pregnant and breastfeeding -- you know, she always made sure that there was a time that I could, you know, go in, and make sure that I could pump … But she made -- she turned the -- this was before they remodeled. She turned the -- what’s it called? The storage closet, that had, you know, all the extra food, and supplies, and stuff -- she turned -- put a rocking chair in there and everything. So she was like, if you’re going to breastfeed, we’re going to make sure you’re comfortable. And she would give me, you know, 20, 30 min to either, you know, let my daughter eat, or pump, and then, you know, I would go back to being in class. And, I mean, she was very -- I don’t feel like I would’ve got -- I don’t feel like I would’ve graduated, I guess you would say.”*

*- Current teenage parent*



*“But as far as the faculty in school, you know, they were really supportive. I mean, they knew that, especially after the baby was born, that I had days where, you know, I would come to school, because I was a parent at that time, my senior year of high school, I was a parent. So, you know, I was going to school, and you know, would be up with the baby all night, I would come into school the next day and end up -- They were pretty flexible with me as far as, you know, if I needed to turn something in late or if I needed to come in late, you know, that sort of thing. They worked with me really well.”*

*- Former teenage parent*



*“[Church members], they said they would like to help out watch my son. And so just after that, just met the church family and they’ve been very supportive. And they’ve just been there for me and my son.”*

*- Former teenage parent*


These tangible supports (e.g., a private place on site for breastfeeding and flexibility with completing assignments) made continuing education possible for these participants. Just as these supports contributed to them seeing the social networks as supportive, the tangible supports also appear to contribute to the teenagers feeling supported as they navigated pregnancy and parenthood.

Not all forms of instrumental support, however, made our participants feel supported. Experiences with social service providers and community organizations led to some pregnant and parenting teenagers feeling stigmatized.


*“I just remember the -- that was the only time I ever felt judged by those ladies that were in [Social Services Clinic] working. They were rude and like so didn’t -- they didn’t seem to care, and it was just like I was coming there for -- I felt like they were already judging me walking through the doors and I never went back. It was a super traumatic experience.”*

*- Former teenage parent*



*“But some of those other programs are the problem like the breastfeeding program. It -- They try to help moms, especially new moms that are breastfeeding, which is great. But, when you fail or if you fail, oh my God, do they make you feel like the lowest of low person out there. I mean, they really do.”*

*- Former teenage parent*


When the pregnant and parenting teens felt judged by the social service providers, they were reluctant to go back to the organizations, creating a barrier to receiving instrumental supports that they may have needed. Concern over being judged and stigmatized by these providers could carry over into the pregnant and parenting teens’ perceptions of other agencies and organizations with whom they have not yet interacted. That is, feeling judged by one provider or at one organization may be generalized and keep the teenagers from seeking necessary instrumental support from other agencies.

### 3.3. Informational Support

Informational support involves learning more about what to expect during pregnancy, birth, and raising a healthy child. Although, the participants wanted to learn about pregnancy and parenting while they were pregnant, they shared experiences of not receiving the information they needed. In particular, our participants discussed wanting more information from healthcare providers.


*“I feel like they could’ve been more -- I feel like, since I was so young, they didn’t really want to explain things to me, because they didn’t want to scare me. But at the same time, they should’ve been raw, and they should’ve been real … I asked about preeclampsia. They wouldn’t really explain that to me. They handed me a pamphlet and said, ‘Here.’”*

*- Current teenage parent*


There was a perceived feeling of judgment from the healthcare providers which impacted their experiences of prenatal care. The current teenage parent went on to explain that she did not believe the pamphlet gave adequate information, and she was left with questions. Stigma and perceived judgment in the healthcare setting became significant barriers to receiving informational support. Perhaps because the informational support from healthcare providers was lacking, the participants turned to other mothers for information about pregnancy and parenting.


*“I’ve been getting advice from a lot of other moms who’ve had multiple children kind of, figuring out what to expect during the labor and the first few months with the babies. And basically, in a sense just kind of feel more ready.*

*- Current teenage parent*


### 3.4. Appraisal Support

Appraisal support can take the form of feedback from others, and it communicates informal and formal expectations. In general, these current and former teenage parents’ stories illustrate the absence of appraisal support. Instead, the participants shared their perceptions that community members did not believe being a pregnant or parenting teenager was acceptable. For example, school faculty, parents, and/or other community members stigmatized their pregnancy and parenthood. Some of the harmful appraisal examples included explicit examples with community members such as them responding negatively to the teenager in public spaces with insulting comments and/or discussing the teenage pregnancy or parenting with others in a negative manner.


*“You know, people would come up to me at Walmart asking me if I knew what condoms were. You know because I was a 17-year-old mom.”*

*- Former teenage parent*



*“Some people like when I see a girl is young and pregnant, some of them—some people will, guess, call them a hoe. Because I guess that they’re not thinking maybe they made a mistake. And they’re kind of alone and they don’t exactly have a support. Especially from the person they got pregnant by or whatever. And so, they’re kind of alone.”*

*- Former teenage parent*


Other participants shared that the negative appraisals were not explicit, but they were communicated more through body language or other types of non-verbal cues.


*“Like you’re really looked down upon. You’re really -- it’s like a disgrace. Because I felt like people -- they were very judgmental. The comment, the look, the stares, the -- like I just felt like, you know, it was just really hard to be pregnant in a smaller town because, you know, everybody first knew about it and then people talked about it and stuff. So it was definitely harder.”*

*- Former teenage parent*



*“That -- I think the school -- I think the principals and the teachers seen that as an embarrassment and disgust that we were pregnant and that you know sex before marriage is a sin and everything. They didn’t want us you know walking around the halls with our big bellies and -- and having other girls think oh that’s cool to be pregnant when it wasn’t.”*

*- Former teenage parent*


### 3.5. Recommendations for Increased Community Support

The participants in this study recommended ways to increase the amount of support for parenting and pregnant teenagers such as increased counseling services, access to contraceptive education, less judgment in social services, and increased resources for teenagers who are parenting.


*“I don’t know if that would be, like, a counseling service or what that would exactly be. But something like that. And something that, you know, they wouldn’t necessarily have to pay to be a part of. I feel like that’s one of the worst thing nowadays is you have to pay for everything. It’s, like, why can’t you have something open just here’s this.”*

*- Former teenage parent*



*“I don’t know if it’s more that’s missing, or it’s just more that’s not talked about. I feel like they could use more information on contraceptives. And even if they choose to have a baby and keep them, generally they could use more information how to care for the baby, and -- -- less judgment*

*- Current teenage parent*


## 4. Discussion

Teenage pregnancy and parenthood have long history of being viewed as social problems in the United States [2] Although teenage pregnancy has been discussed and examined over the years, progress is needed in finding effective ways to provide support to pregnant and parenting teenagers [22,23]. We contend that framing teenage pregnancy and parenthood as social problems adversely affects the social support that pregnant and parenting teens receive in their families, schools, and communities. This is concerning as having higher quality relationships within the teenagers’ support networks provides crucial support for the pregnant and parenting teenagers [21]. We did not ask the participants specific questions regarding their mental health status, however, many participants shared positive feelings while receiving instrumental and emotional support. Negative appraisal support or a lack of support was reported as making the participants feel bad about themselves or negatively affecting their mental health.

Our results can provide insights into where pregnant and parenting teenagers find helpful social support, and unfortunately where they do not, and aid in structuring effective support systems for rural counties. As it can be seen in Figure 2, these current and former teenage parents had mixed experiences in the emotional and informational support domains, with them having positive experiences in the instrumental domain, and negative experiences in the appraisal domain. By unpacking the sources, types, and functions of pregnant and parenting teenagers’ social support, our results can help to inform the development of family, school, and community resources to support pregnant and parenting teenagers.

### 4.1. Emotional Support

These current and former teenage parents described how a variety of relationships (e.g., parents, caregivers, stepparents, friends, teachers, and church family) formed their emotional support network. The wide range of relationships where our participants found emotional support aligns with the prior research, illustrating that emotional support from parents is crucial, but emotional support in other domains is also impactful [25]. Within these networks, our participants had a mixed experience with emotional support. The perceptions of positive emotional support appeared to be closely tied with tangible rituals that are commonly associated with pregnancy and parenthood. For example, some parents, friends, and churches celebrated our participants’ pregnancies by hosting baby showers and parties.

It is also important to note that not all of the parents were initially accepting of their children’s pregnancies. Initially not accepting their children’s pregnancies created a context in which the emotional support from the parents was absent. For example, the participants reported being kicked out of their home, disowned, given the silent treatment, and other forms of disapproval from family members in the early days of their pregnancy, which caused emotional distress and impacted their prenatal health and how the teenagers responded to their parenting responsibilities. Fortunately for some of our participants whose parents were not initially supportive, they did come to accept their children’s decision to parent and began providing emotional support. This suggests that support is fluid within the social support theory, and the level of it may increase and decrease throughout the pregnancy and parenting processes.

### 4.2. Instrumental Support

Pregnant and parenting teens can receive instrumental support within their families, from professionals such as healthcare providers and teachers, and community organizations [21]. For some of our participants, instrumental support within their schools was critical for them to be able to complete their educations. In this context, instrumental support involved being flexible and understanding regarding the submission of assignments, providing a private space for teenage mothers to pump breast milk or feed their babies, and one school district provided free daycare for infants and toddlers onsite. The pregnant and parenting teenagers who had access to the onsite daycare attest that this resource was one of the primary reasons they were able to continue their education and earn their diplomas. There is a correlation between earning a high school diploma and an increase in positive health outcomes [26]. Conversely, a correlation exists between teenage pregnancy and decreased rates of completing a degree in a mainstream high school [27] or completing high school at all [28]. These outcomes show that the instrumental support provided by the local schools can contribute to assisting the pregnant and parenting teenagers in achieving their educational goals.

Although our participants recounted positive instrumental support from some teachers and school officials, they did not perceive the available healthcare and social service resources within the communities to be a comfortable or non-judgmental environment to receive instrumental support. The participants reported feeling stigmatized or a lack of patience or empathy from their providers. The lack of instrumental support from the community emerged through various means, whether it was due to a bias towards their age, the teenagers’ ability to be a parent, or general judgment of their decisions. This finding of teenagers experiencing bias and judgment from community members is in alignment with previous findings [2,29].

### 4.3. Informational Support

Our results indicate that pregnant teenagers do not receive adequate informational support from family members, professional healthcare, or social service providers. That is, pregnant and parenting teens may not be learning what they think they need to know about pregnancy and parenthood. It is possible that the judgment and stigma perceived by the participants were barriers to receiving medical information from healthcare providers. It is also possible that healthcare providers had a bias toward the young parent and did not allot the appropriate time to provide information and education about the parents’ and child’s health. Many rural areas are considered to be medical care deserts due to them not having enough social services and healthcare providers. This deficit, in combination with an ongoing stigma around teenage pregnancy and other personal health issues in rural counties, is related to the lack of informational support offered to teenagers [11].

### 4.4. Appraisal Support

Appraisal support is information that drives self-evaluation, which in turn can affect self-esteem. These current and former teenage parents recalled experiences of being stigmatized. This was especially common among these young mothers who recalled being described as a “hoe”, “slut”, or “a girl with loose morals”. Experiencing negative appraisal at the community level is not uncommon for teenage parents [12,21] Judgement within their communities seems to communicate to these current and former teenage parents that their experience was not acceptable. The feedback of negative appraisal caused increased stress by them not feeling welcome in their schools or community, which made the positive emotional support even more important and valuable. This can lead to pregnant and parenting teenagers isolating themselves, and in turn, this can lead to them not receiving the help and medical care that they need [12]. Therefore, that absence of appraisal support may result in pregnant and parenting teens internalizing the community stigma, creating barriers to accessing other forms of social support in their communities.

### 4.5. Recommendations for Social Support

The participants were asked to share the resources they wished were in their communities that addressed their needs. Many of them reported more counseling and mental health services as resources for prevention and intervention. Additionally, the participants expressed the desire for more services that focused on their lived experiences related to trauma, professionals trained in adolescent health, educational opportunities for the community to become less judgmental towards teenagers, and safe places that are teenage-friendly. These characteristics have been cited as best practices for social services and healthcare services [2].

### 4.6. Implications

Multifaceted social support is critical for promoting the well-being of parenting teenagers and their children. Our results provide insights into public health interventions for pregnant and parenting teens. Firstly, ensuring that pregnant and parenting teens have access to tangible, instrumental supports may be key to promoting other forms of social support (e.g., emotional and appraisal support). Our participants described two forms of instrumental support that could become public health interventions: (1) baby showers and (2) education flexibility. Baby showers appear to communicate to pregnant and parenting teenagers that they are being supported. Therefore, community organizations could make baby showers part of their outreach efforts. Further, it is not uncommon for rural towns and cities to hold community “baby showers” as broad outreach efforts to show support beyond the pregnancy into parenthood. Within those events, specific resources could focus on the needs of the pregnant and parenting teenagers, as well as ensuring that the teenagers feel welcome. A second area of intervention should focus on providing pregnant and parenting teenagers with necessary flexibility so they can continue their education. In addition to providing flexibility related to completing assignments, pregnant and parenting teens, especially mothers, appear to benefit when they have supports related to childcare and feeding. We recognize that not all high schools are equipped to be able to offer these types of support. Therefore, community agencies could create resources for high schools with concrete suggestions for how they can support pregnant and parenting students. The high schools could use those resources to create support plans with the primary goal of earning a high school diploma. The support plans could include creating comfortable and private places to pump breastmilk and feed their babies onsite, contingency plans for absences, emotional support check-ins with a counselor or teacher, and partnerships with health care providers for medical care. These types of supports need to be available for the father too.

Finally, the absence of informational support among our participants may indicate that healthcare and social services providers need additional training on the best practices for a more youth-centered approach. Creating effective and respectful partnerships with pregnant and parenting teenagers (e.g., shared decision-making, respectful and non-judgmental communication, and clear expectations, etc.) can increase positivity in the maternity experience and outcomes [30].

### 4.7. Limitations

The limitations of the present study must be considered when one is interpreting the results. The research team partnered with two specific rural counties in a Midwestern state. This study was small in scope; thus the participants’ encounters do not reflect the experiences of all pregnant and parenting teenagers in all rural communities in the United States, therefore, the findings are not generalizable to other similar rural populations. The interview guides and codebooks were created with these county demographics and characteristics in mind and their unique needs which were identified during the relationship-building phase. Another limitation is that most of the participants were female. This could be because females bear most of the responsibility while they are pregnant, therefore, they had an increased interest to participate in this study because they had more experiences to share. Future studies should examine rural male teenage parents’ perceptions of the social support provided in their community and explore what types of supports would be beneficial. Additionally, future studies should collect data on obstetric characteristics (e.g., pregnancy trimester and/or time passed since birth), as well as additional sociodemographic characteristics (e.g., educational level attained), as we did not collect these data in our specific study.

Most of the participants from one county in this study were recruited through a coalition that was already working to address teenage pregnancy and had been in communication about changing teenage pregnancy norms within their community. There is a potential for bias if the participants in the study included individuals who were already interested in the topic of teenage pregnancy. Their contributions to the study could potentially be different from those of the individuals who did not enroll due to a lack of interest and awareness or availability.

## 5. Conclusions

Identifying specific strategies to provide emotional, instrumental, informational, and appraisal support to create positive outcomes for all pregnant and parenting teenagers, as well as their children, will take the collective effort of all community members, including teenagers, youth workers, healthcare providers, and policymakers. Although there are some social supports available to rural teenagers who are pregnant and/or parenting, there needs to be more attention on providing explicit emotional, instrumental, informational, and appraisal support systems. It is crucial for the rural community social services to collaborate and provide comprehensive resources to support a healthy pregnancy and effective parenting skills for the teenagers. A comprehensive approach includes parents, school personnel, health care providers, social service providers, and key community stakeholders such as policymakers to increase physically and emotionally safe, teen-friendly resources and services. Collaboration is an initial strategy to fully support the needs of these teenagers and decrease the stigma towards pregnant and/or parenting teenagers.

## Figures and Tables

**Figure 1 ijerph-19-16906-f001:**
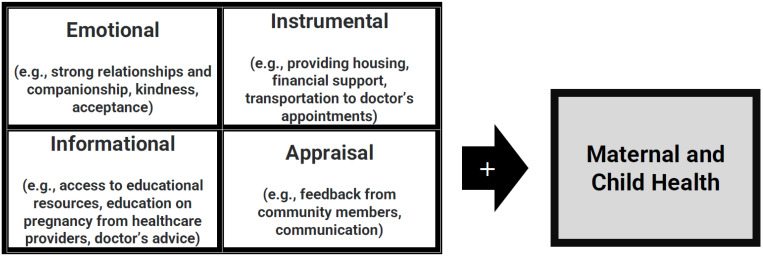
Conceptual framework based on House’s theory of social support (House, 1981).

**Figure 2 ijerph-19-16906-f002:**
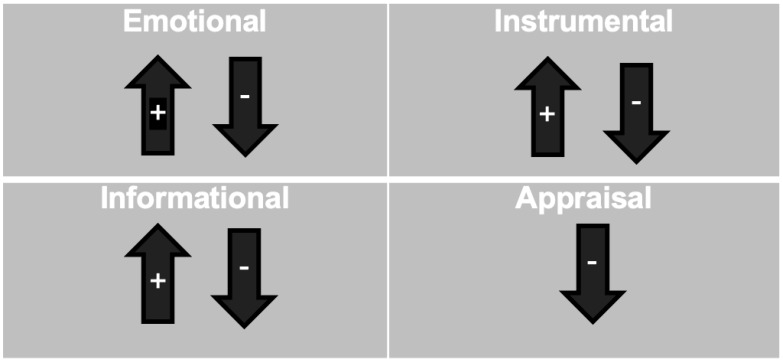
Positive vs. negative support reported.

## Data Availability

The study data are available upon request from the corresponding author, subject to appropriate ethical review by the IRB of the author’s institution.

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
