# Peer review of "“It Was Just Really Hard to Be Pregnant in a Smaller Town …”: Pregnant and Parenting Teenagers’ Perspectives of Social Support in Their Rural Communities"

_ijerph, 2022, doi:10.3390/ijerph192416906_

Round 1

Reviewer 1 Report

Overall, excellent article covering an important topic about health and healthcare in rural communities. I have a few suggestions/comments below:

Introduction: 

- Please clarify what you mean by the first sentence of section 1.1. 

Materials and methods: 

- Please provide more information about how participants were recruited. When participants found out about the study, how did they come to be able to participate? Were healthcare providers involved in directly helping to select participants? Were participants male, female, either? Why were some adults included? What was the breakdown of teenagers vs adults? What age ranges were you using to classify each?

- Consider including the questions you used in an appendix. 

- Please include 2.2 in the results section since it includes information about the participants instead of materials & methods. 

- How big was the research team? Who were the people invovled? 

Results:

- Great highlights of important quotes that touch on some important elements of being a teenage parent in rural communities 

Discussion:

- Please include citations for the first sentence of the first paragraph in the discussion 

- I particularly liked the implications section, great suggestions provided here 

Author Response

Reviewer 1

Thank you, reviewer 1, for your thoughtful comments and suggestions. Below I have responded to each comment.

  1. Please clarify what you mean by the first sentence of section 1.1. 

We have rewritten the first sentence to more clearly state the teen birth rate in the United States. It now reads, “In 2021, the United States had a teenage birth rate of 14.4 births per 1,000 15-19 year-old women (Hamilton, Martin & Osterman, 2022).”

  1. Please provide more information about how participants were recruited. When participants found out about the study, how did they come to be able to participate?

We have provided more information about how participants were recruited (page 3, lines 121-126). Briefly, participants were identified by having members of community coalitions, social service agencies, schools, and faith-based organizations nominate current or former teen parents to participate in our project. We asked everyone nominated if they wanted to be interviewed. We also placed flyers advertising the opportunity to be interviewed in social service and healthcare offices. Finally, we asked each person we interviewed if they knew of anyone, we should invite to be interviewed.

  1.  

3a Were healthcare providers involved in directly helping to select participants?

There were healthcare providers who allowed us to place flyers in their offices to aid in the recruitment process. Additionally, we worked with one healthcare provider who sent letters developed by our research team to patients who were current or recent pregnant teens.

A sentence was added (page 3, lines 126-127) Second, an obstetrician sent letters to current and recent pregnant teens.

3b Were participants male, female, either?

We included participants who identified as male and female.

3c. Why were some adults included?

Adults who experienced parenthood as teens were included in the study. While the target communities have high teenage pregnancy rates, they are very small, rural populations. Thus, the actual pool of current pregnant and parenting teens is limited. However, teenage pregnancy and parenthood have been long-standing issues in these communities, thus  we determined that including participants who were not currently teenagers, but had been pregnant or parenting as teenagers, would provide meaningful insights into the experiences of being a pregnant or parenting teen in these rural communities. Therefore, we included persons who were not teenagers at the time of their interviews to participate in our project. All the “former” teenage parents in our sample were living in the communities when they were pregnant and parenting as teenagers.

3d What was the breakdown of teenagers vs adults? What age ranges were you using to classify each?

The research team were identified as were aged 15-19. We spoke to 4 participants who were in their teenage years and pregnant. 22 of the participants were 20 years or older.  This information is included in the participants section, now read in the results.

Page 4 ‘Results’ section beginning on the top of page 5 lines 267-264 reads, “A total of 26 current and former pregnant and/or parenting teenagers participated in this study. All participants lived in the partnering counties during their pregnancy and became pregnant or parents between the ages of 15-19 years old. Four participants were currently pregnant teenagers at the time of the interview and were all 18- or 19-years-old. Twenty-two participants had children as teenagers and were over19 years old at the time of the interview. Most participants identified as female (n = 23), and three participants identified as male. “

  1. Consider including the questions you used in an appendix. 

Thank you for the suggestion.

  1. Please include 2.2 in the results section since it includes information about the participants instead of materials & methods. 

Thank you for the suggestion. The participants section has been moved to the beginning of the results section.

  1. How big was the research team? Who were the people involved? 

The  research team included undergraduate (1) and graduate students (6), faculty members (3), community health professionals (1), and leaders from community organizations (2).

  1. Great highlights of important quotes that touch on some important elements of being a teenage parent in rural communities 

Thank you!

  1. Please include citations for the first sentence of the first paragraph in the discussion 

A citation has been added to the end of the sentence.

  1. I particularly liked the implications section, great suggestions provided here 

Thank you!

Reviewer 2 Report

Dear Authors

Attached is a review of your work, which I generally rate highly

Kind regards

Reviewer

Author Response

Thank you, Reviewer 2, for your time and thoughtful feedback.